# Brain States as Control Signals for Next-Token Distributions in Language Models

## Abstract

Biological cognition depends on latent neural states, yet language models are typically conditioned on text alone. We ask a mechanistic question: can a brain-state vector act as a *structured control input* that predictably modulates a language model's next-token distribution? Using paired text contexts and simulated multi-region brain activity, we train brain-conditioned autoregressive LMs and quantify influence via Jensen-Shannon divergence (JSD) and next-token negative log-likelihood (NLL). Across brain dimensionalities $d \in \{68, 136, 272, 544\}$, brain conditioning induces progressively stronger deformation of the next-token distribution relative to a zero-brain baseline. This effect is not a trivial perturbation: shuffled and Gaussian-matched brain vectors produce similar deformation but degrade NLL, while real brain vectors consistently improve NLL, alternatively freezing the brain pathway eliminates all effects. Together, these results establish a quantitative framework for neural-state control of language generation that distinguishes generic distributional perturbations from task-aligned control.

## 1. Introduction

Large language models (LMs) generate text by predicting the next token from preceding context, effectively treating language as a purely text-conditioned process. Biological cognition, in contrast, is shaped by continuously evolving neural states reflecting perception, memory, affect, and internal dynamics. This raises a natural modeling question: can a latent neural state serve as a *control signal* that modulates linguistic prediction?

Prior work in brain-computer interfaces (BCI) has established that neural activity contains decodable information about intended communication, including handwriting and speech, using invasive recordings (Anumanchipalli et al., 2019; Makin et al., 2020; Willett et al., 2021; Moses et al., 2021; Willett et al., 2023). More recently, non-invasive recordings have been used to reconstruct aspects of continuous language semantics from fMRI (Tang et al., 2023). While these efforts primarily emphasize *decoding* language from neural measurements, our focus is different: we treat a brain-state vector as an *exogenous input* to a language model and ask whether it exerts measurable, structured control over the model's next-token distribution.

A second relevant thread is controllable generation. Techniques such as attribute control and plug-and-play steering demonstrate that relatively low-dimensional signals can shift model behavior (Keskar et al., 2019; Dathathri et al., 2020), and guidance methods in generative modeling formalize continuous "control knobs" for distribution shaping (Dhariwal & Nichol, 2021). However, output distributions can be substantially altered without improving next-token likelihood. This motivates a mechanistic evaluation that separately measures (i) the magnitude of distributional deformation induced by control and (ii) whether this deformation increases probability mass on the correct continuation.

In this work, we study brain-conditioned language modeling under paired data of text contexts and multi-region brain-state vectors. Rather than evaluating generation quality subjectively, we quantify brain influence by directly comparing next-token distributions with and without brain input. For a context $x_{1:t}$ and brain state $z$, we measure deformation via Jensen-Shannon divergence between $p(\cdot \mid x_{1:t}, z)$ and a zero-brain baseline $p(\cdot \mid x_{1:t}, 0)$, and we measure utility via next-token negative log-likelihood (NLL) on the ground-truth token. This framework explicitly distinguishes generic perturbations from control signals that move probability mass toward correct predictions.

**Contributions.** (i) We formalize a brain-conditioned autoregressive LM setup in which a multi-region brain state is injected into a Transformer (Vaswani et al., 2017) via a lightweight learned projection, analogous in spirit to feature modulation and parameter-efficient conditioning (Perez et al., 2018; Houlsby et al., 2019; Hu et al., 2021). (ii) We introduce a paired evaluation of *deformation* (JSD, top-1 agreement) and *utility* ($\Delta$NLL) to quantify whether brain input provides structured, predictive control rather than arbitrary distribution shift. (iii) Across brain dimensionalities,

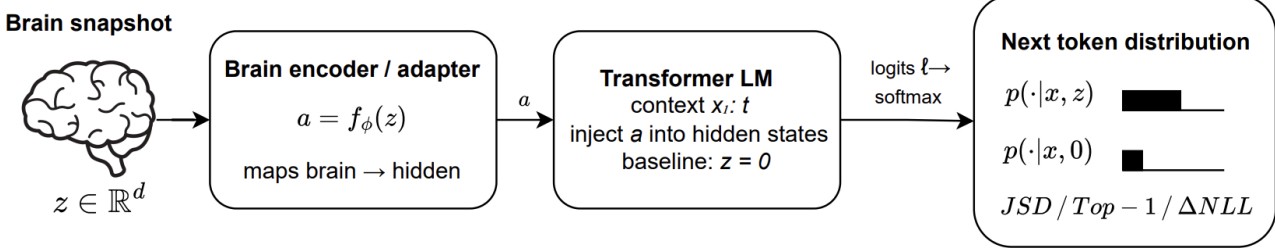

*Figure 1.* Brain-conditioned language modeling as controllable next-token distribution deformation. A brain snapshot $z$ is encoded into a control vector $a$ and injected into a Transformer LM's hidden states. We compare the resulting next-token distribution $p(\cdot \mid x, z)$ against the zero-brain baseline $p(\cdot \mid x, 0)$ using Jensen-Shannon divergence (JSD), top-1 agreement, and $\Delta$NLL.

we show monotonic increases in deformation strength and validate causality with shuffled/Gaussian controls and a frozen-pathway intervention. (iv) We further characterize the *content* and *geometry* of control using token-category sensitivity and logit-space PCA, probing whether control concentrates on semantic content and whether it operates through a compact subspace.

## 2. Brain-Conditioned Language Modeling and Evaluation

Let $x_{1:t}$ denote a sequence of tokens and $z \in \mathbb{R}^d$ a brain-state vector representing activity across $d$ regions. We train an autoregressive language model parameterized by $\theta$ to predict the next token conditioned on both text and brain state:

$$p_\theta(x_{t+1} \mid x_{1:t}, z) = \text{softmax}(f_\theta(x_{1:t}, z)). \quad (1)$$

The model is trained using standard next-token cross-entropy:

$$\mathcal{L}(\theta) = -\sum_t \log p_\theta(x_{t+1} \mid x_{1:t}, z). \quad (2)$$

Brain conditioning is implemented via a learned projection that injects $z$ into the transformer hidden states. When $z = \mathbf{0}$, the model reduces to a standard text-only LM.

### 2.1. Measuring Brain Influence

For a fixed context $x_{1:t}$, we compare:

$$p = p_\theta(\cdot \mid x_{1:t}, z), \quad (3)$$
$$q = p_\theta(\cdot \mid x_{1:t}, \mathbf{0}). \quad (4)$$

We measure their divergence using Jensen-Shannon divergence:

$$\text{JSD}(p \parallel q) = \frac{1}{2}\text{KL}(p \parallel m) + \frac{1}{2}\text{KL}(q \parallel m), \quad m = \tfrac{1}{2}(p+q). \quad (5)$$

We additionally report top-1 agreement, the fraction of samples for which $\arg\max p = \arg\max q$. Together, these metrics quantify how strongly the brain state alters token-level predictions.

### 2.2. Data

We construct paired datasets of $(x_{1:t}, z, x_{t+1})$ triples by combining text from a fixed general-domain corpus derived from Wikipedia (Wiki-40B; following (Raffel et al., 2020)) with simulated multi-region brain activity generated using The Virtual Brain (TVB) (Schirner et al., 2022). We extract brain snapshots $z \in \mathbb{R}^d$ from $d$ cortical regions at matched time points. We build separate datasets for brain dimensionalities

$$d \in \{68, 136, 272, 544\},$$

each containing approximately 100k samples. For each $d$, a separate LM is trained to ensure architectural compatibility.

## 3. Experiments

We evaluate whether brain snapshots act as a structured control input for next-token prediction, and whether observed distribution shifts reflect meaningful brain-text alignment rather than generic perturbation. Unless noted otherwise, all metrics are computed on 5,000 held-out contexts.

### 3.1. Control strength scaling with brain dimensionality

We test whether higher-dimensional brain snapshots yield stronger modulation of the model's next-token distribution. For each $d \in \{68, 136, 272, 544\}$, we train a brain-conditioned LM using the same text corpus, architecture family, and optimization settings, varying only the brain snapshot dimensionality $z \in \mathbb{R}^d$ and the corresponding brain-to-hidden projection.

To quantify control strength, we measure how much the next-token distribution changes when conditioning on a real

*Table 1.* Brain influence increases with brain dimensionality. JSD is computed between next-token distributions with real brain input vs. zeroed brain.

| Brain dim | JSD mean | JSD median | JSD std | Top-1 agree |
|-----------|----------|------------|---------|-------------|
| 68 | 0.0657 | 0.0511 | 0.0604 | 0.681 |
| 136 | 0.0736 | 0.0540 | 0.0696 | 0.660 |
| 272 | 0.0857 | 0.0718 | 0.0710 | 0.620 |
| 544 | 0.1359 | 0.1162 | 0.1073 | 0.577 |

*Table 2.* Controls for the 136-D brain-conditioned model. Real brain conditioning improves next-token likelihood despite inducing distributional deformation comparable to shuffled and Gaussian controls.

| Condition | JS mean | Top-1 agree | NLL mean | $\Delta$NLL vs. ZERO |
|-----------|---------|-------------|----------|----------------------|
| REAL | 0.0737 | 0.661 | 2.417 | $-0.376$ |
| ZERO | - | 1.000 | 2.793 | 0.000 |
| SHUF | 0.0756 | 0.655 | 2.931 | $+0.138$ |
| GAUSS | 0.0739 | 0.654 | 2.932 | $+0.139$ |

brain snapshot, relative to a zero-brain baseline. Concretely, for each $d$ we compute Jensen-Shannon divergence (JSD) between $p(\cdot \mid x_{1:t}, z)$ and $p(\cdot \mid x_{1:t}, \mathbf{0})$, along with top-1 agreement as a measure of argmax stability.

Table 1 summarizes these metrics across dimensionalities. Mean JSD increases monotonically with $d$ ($0.066 \rightarrow 0.136$), while top-1 agreement decreases ($0.681 \rightarrow 0.577$), indicating that richer brain representations produce larger and more frequent shifts in token probabilities relative to the text-only baseline.

### 3.2. Controls: separating generic perturbation from aligned signal

A central challenge in evaluating external control signals is that distributional change alone is not informative. Injecting almost any additional vector into a language model can deform logits, alter the argmax token, and increase divergence from a baseline-without improving prediction. To claim that brain snapshots provide meaningful control, we must therefore show that their effects cannot be replicated by unstructured or misaligned perturbations.

**Control conditions.** For each evaluation context, we consider four conditioning regimes: (i) **REAL**, the true brain vector paired with the text; (ii) **ZERO**, an all-zero brain vector; (iii) **SHUF**, brain vectors randomly permuted across samples, preserving marginal statistics but breaking alignment; (iv) **GAUSS**, Gaussian vectors matched to the empirical mean and variance of the brain dataset.

For each condition, we compare the resulting next-token distribution to the ZERO baseline. We report (a) Jensen-Shannon divergence (JSD) as a measure of distributional deformation, (b) top-1 agreement as a measure of argmax

stability, and (c) next-token negative log-likelihood (NLL) on the ground-truth token. Changes in predictive performance are summarized as $\Delta$NLL relative to ZERO.

Table 2 reveals a clear dissociation between deformation magnitude and predictive usefulness. All non-zero conditioning vectors-REAL, SHUF, and GAUSS-induce comparable Jensen-Shannon divergence relative to ZERO, confirming that substantial distributional deformation is easy to obtain. Likewise, all three reduce top-1 agreement to a similar extent, indicating frequent changes to the argmax token.

Crucially, only the REAL brain condition improves next-token prediction. Conditioning on the true brain vector reduces NLL by $0.376$, while both SHUF and GAUSS *worsen* prediction despite inducing nearly identical distributional shifts. This demonstrates that the benefit of brain conditioning cannot be attributed to generic perturbation, added variance, or increased expressivity alone. Instead, predictive gains depend on structured alignment between the brain state and the linguistic context.

**Frozen pathway control.** To verify that these effects are mediated by the learned brain-conditioning mechanism itself, we perform an additional causal control by freezing the brain projection to zero. Under this intervention, all conditioning regimes collapse to the ZERO baseline: Jensen-Shannon divergence drops to zero, top-1 agreement returns to one, and NLL becomes identical across REAL, SHUF, and GAUSS. This confirms that both distributional deformation and predictive gains arise specifically from the learned brain injection pathway, rather than from incidental architectural or evaluation artifacts.

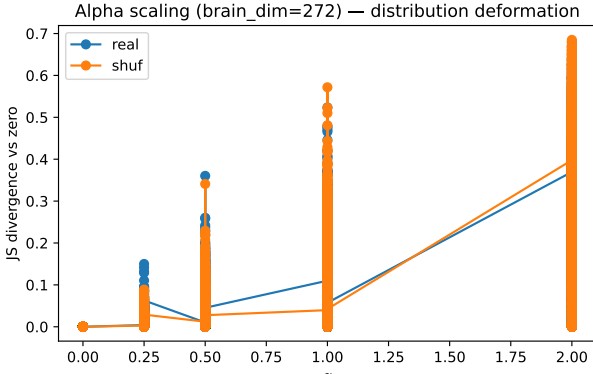

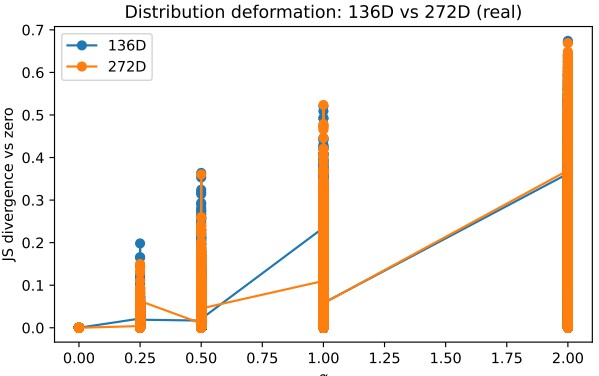

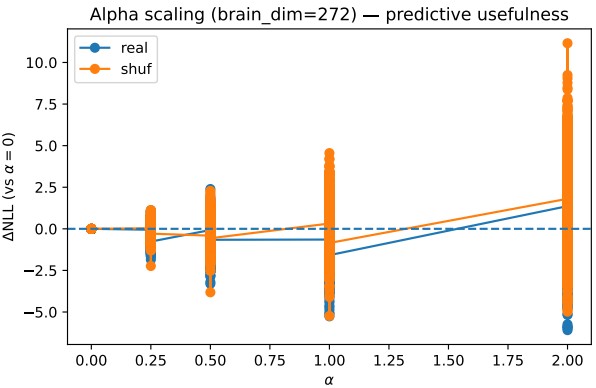

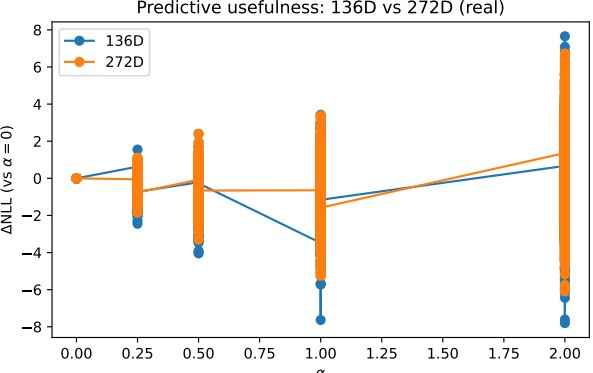

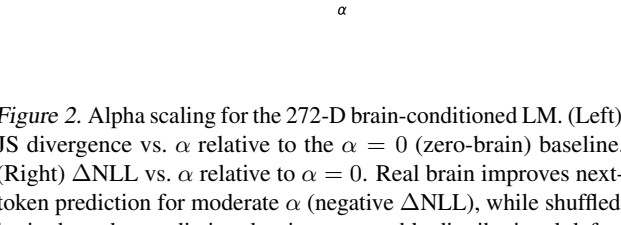

*Figure 2.* Alpha scaling for the 272-D brain-conditioned LM. (Left) JS divergence vs. $\alpha$ relative to the $\alpha = 0$ (zero-brain) baseline. (Right) $\Delta$NLL vs. $\alpha$ relative to $\alpha = 0$. Real brain improves next-token prediction for moderate $\alpha$ (negative $\Delta$NLL), while shuffled brain degrades prediction despite comparable distributional deformation.

*Figure 3.* Direct comparison of alpha scaling between 136-D and 272-D models (REAL condition). Overlaying curves on identical axes reveals differences in deformation strength (JS) and predictive usefulness ($\Delta$NLL) that can be visually subtle in separate plots.

### 3.3. Dose-response: scaling the brain signal

To test whether brain conditioning behaves like a graded control knob, we scale the brain vector $z \leftarrow \alpha z$ with $\alpha \in \{0, 0.25, 0.5, 1, 2\}$. For each $\alpha$, we report JSD and $\Delta$NLL relative to $\alpha = 0$.

For both REAL and SHUF, JSD typically increases with $\alpha$, indicating smooth growth in deformation magnitude. However, REAL and SHUF differ in predictive effect: REAL tends to reduce NLL for moderate $\alpha$, while SHUF degrades NLL despite comparable deformation.

Figure 2 shows the same pattern in the 272-D model: deformation grows with $\alpha$ for both conditions, while predictive gains are specific to REAL. Figure 4 shows that this qualitative dissociation persists at 544-D: both REAL and SHUF induce smoothly increasing deformation with $\alpha$, but predictive gains remain stronger (and degradation at large $\alpha$ is less severe) for REAL than for SHUF.

### 3.4. Token-level sensitivity: semantic vs. syntactic control

To assess whether brain conditioning preferentially modulates semantic content rather than surface-level structure, we perform a lightweight token sensitivity analysis. For each evaluation context, we compute the absolute logit change induced by brain conditioning,

$$|\Delta\ell| = |\ell(x \mid z) - \ell(x \mid 0)|,$$

and group vocabulary items into three coarse linguistic categories: (i) punctuation, (ii) stopwords (function words), and (iii) content tokens (nouns, verbs, adjectives, numerals).

Table 3 reports mean $|\Delta\ell|$ for each token category under real, shuffled, and Gaussian brain inputs in the 136-D model. Across all conditions, content tokens exhibit consistently larger logit changes than stopwords (content/stopword ratio $\approx 1.11\times$), indicating that brain conditioning preferentially affects semantically meaningful tokens. Punctuation tokens show comparable but not dominant sensitivity, suggesting



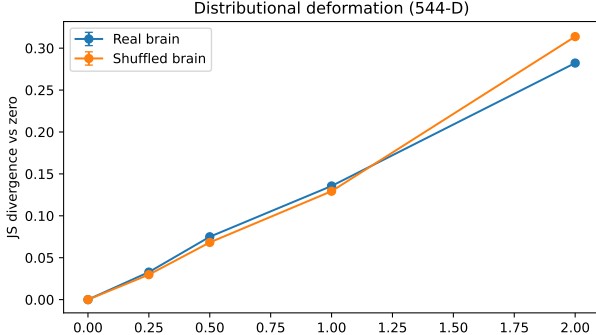

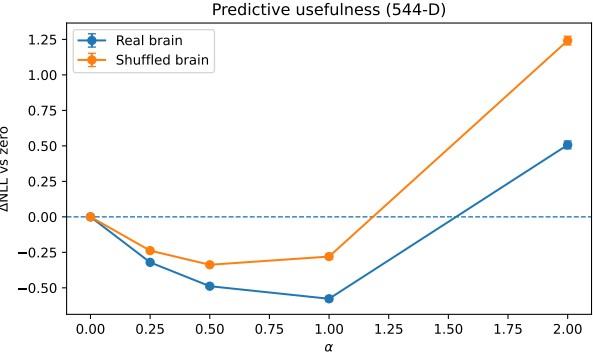

*Figure 4.* Alpha scaling for the 544-D brain-conditioned LM. (Left) JS divergence vs. $\alpha$ relative to the $\alpha = 0$ (zero-brain) baseline. (Right) $\Delta$NLL vs. $\alpha$ relative to $\alpha = 0$. Both real and shuffled brain vectors induce smoothly increasing deformation, but real brain yields stronger predictive gains at moderate $\alpha$ and substantially less degradation at large $\alpha$.

that deformation is not driven purely by syntactic or formatting effects.

Crucially, although shuffled and Gaussian brain inputs induce nearly identical token-category sensitivity profiles to real brain inputs, only the real brain condition improves next-token likelihood (Section 3.2). This dissociation demonstrates that semantic-weighted deformation alone is insufficient: predictive gains arise specifically from structured alignment between brain state and linguistic context.

**Scaling with brain dimensionality.** We next examine how token sensitivity evolves as brain dimensionality increases. Table 4 aggregates mean $|\Delta\ell|$ by token category for real brain conditioning across the 136-D, 272-D, and 544-D models.

As dimensionality increases, content-token sensitivity grows substantially, while punctuation sensitivity decreases slightly and stopword sensitivity increases more modestly. This widening gap indicates that higher-dimensional brain representations increasingly concentrate deformation on se-

mantically meaningful tokens. In the 272-D and 544-D models, content words dominate logit deformation, whereas punctuation tokens remain comparatively stable.

**Frozen brain pathway control.** To verify that the observed effects are mediated specifically by the learned brain-conditioning pathway (rather than incidental architectural differences), we repeat the control evaluation with the brain projection frozen to zero, effectively disabling brain injection while keeping the rest of the model unchanged. Under this intervention, all conditioning regimes (REAL, SHUF, GAUSS) become indistinguishable from the ZERO baseline: Jensen-Shannon divergence collapses to 0.000 (mean/median/std all 0), top-1 agreement becomes 1.000, and the next-token negative log-likelihood is identical across conditions (NLL mean 2.8369 for REAL/ZERO/SHUF/GAUSS over 5,000 samples). This confirms that both distributional deformation and predictive gains arise causally from the learned brain projection, rather than from generic conditioning artifacts or evaluation noise.

Together, these results show that brain conditioning does not induce uniform or syntactic perturbations. Instead, as brain dimensionality increases, deformation becomes increasingly concentrated on content words, indicating semantic control. When combined with the negative log-likelihood improvements observed only for real brain inputs, this provides direct evidence that brain-state vectors act as a structured semantic control signal over the language model's output distribution.

### 3.5. Logit-space geometry: low-rank control and directional alignment

To characterize the geometry of brain-induced control, we analyze the logit deformation using principal component analysis (PCA) over 5,000 contexts and a fixed 2,000-token vocabulary subset.

We first condition on contexts where real brain input improves prediction ($\Delta$NLL$_{\text{real}} < -0.1$; 2,791 samples) and compare PCA spectra for (i) REAL brain input and (ii) SHUF brain input applied to the same contexts. We additionally analyze SHUF deformations on contexts where shuffled input is harmful ($\Delta$NLL$_{\text{shuf}} > 0.1$; 2,108 samples).

Across all conditions, logit deformation is strongly low-dimensional. In the 272-D model, the first principal component explains $\approx 31\%$ of variance, the first 10 components explain $\approx 53\%$, and the effective rank is approximately 8 (Table 5). This demonstrates that brain conditioning acts through a compact control subspace in logit space rather than inducing diffuse perturbations across the vocabulary.

Notably, the variance spectra and effective rank are nearly identical for REAL and SHUF deformations, both on helpful

*Table 3.* Token-level sensitivity to brain conditioning in the 136-D model. We report mean absolute logit change $|\Delta\ell|$ for three token categories under real, shuffled, and Gaussian brain inputs.

| Condition | Punctuation | Stopwords | Content | Content / Stop | Content / Punct |
|---|---|---|---|---|---|
| REAL | 0.891 | 0.731 | 0.815 | 1.115 | 0.961 |
| SHUF | 0.900 | 0.752 | 0.834 | 1.109 | 0.971 |
| GAUSS | 0.900 | 0.750 | 0.835 | 1.115 | 0.971 |

*Table 4.* Mean absolute logit change $|\Delta\ell|$ by token category for real brain conditioning. Higher-dimensional brain representations increasingly target semantic content.

| Brain dim | Punctuation | Stopwords | Content words |
|---|---|---|---|
| 136 | 0.021 | 0.034 | 0.041 |
| 272 | 0.018 | 0.039 | 0.062 |
| 544 | 0.015 | 0.044 | 0.089 |

and harmful contexts. This indicates that predictive usefulness does not arise from lower-rank or more concentrated deformation. Instead, REAL and SHUF induce geometrically similar low-dimensional deformations, but only REAL aligns these deformations with directions that increase probability mass on the correct token.

These results show that brain conditioning defines a low-dimensional control subspace, while predictive gains arise from directional alignment with the task rather than from deformation magnitude or rank.

## 4. Discussion

This work addresses a mechanistic question about controllable language generation: whether a latent brain-state vector can function as a structured control signal that predictably modulates a language model's next-token distribution. Across a range of analyses, we find consistent evidence that brain conditioning induces systematic, graded, and partially semantic deformation of token probabilities, rather than arbitrary perturbation.

A central result is that control strength scales with the dimensionality of the brain representation. As brain dimensionality increases from 68 to 544, we observe monotonic growth in distributional deformation (Jensen-Shannon divergence) alongside reduced argmax stability. This pattern suggests that higher-dimensional brain states provide a richer set of effective control directions in the model's hidden and logit spaces, enabling more frequent and larger shifts in next-token probabilities. Crucially, this scaling behavior would be unlikely if the brain input were treated as noise or ignored by the model.

At the same time, our results show that distributional defor-

mation alone is insufficient to characterize useful control. Shuffled and Gaussian-matched brain vectors induce deformation magnitudes comparable to real brain input, yet consistently degrade next-token likelihood. Only real brain conditioning shifts probability mass toward the correct continuation. This dissociation highlights the importance of separating *how much* a control signal changes the output distribution from *whether* that change is aligned with the task.

Beyond magnitude, we find evidence that brain conditioning is not uniform across the vocabulary. Token-level sensitivity analyses show that deformation increasingly concentrates on content words as brain dimensionality grows, while punctuation and function words are comparatively less affected. Although shuffled controls exhibit similar sensitivity profiles, only real brain input yields predictive gains, suggesting that semantic-weighted deformation becomes useful only when it is aligned with the linguistic context. This points to a notion of *semantic control*: brain states bias the model toward certain meanings or concepts rather than merely perturbing the surface.

Finally, analysis of logit-space geometry reveals that brain-induced deformation operates through a compact, low-dimensional subspace. Both real and shuffled brain inputs produce low-rank changes with similar variance spectra, indicating that predictive usefulness does not arise from stronger or more concentrated deformation. Instead, usefulness depends on *directional alignment* within a shared control subspace. Taken together, these findings suggest that brain conditioning acts as a continuous control knob in logit space: its magnitude determines how far the distribution moves, while alignment determines whether the movement improves prediction.

More broadly, this work reframes brain-conditioned language modeling as an interpretable control problem. Rather than asking whether language can be decoded from neural activity, we ask how external latent states influence probabilistic language prediction. This perspective reframes neural control as a problem of distributional geometry and causal intervention on next-token probabilities.

*Table 5.* PCA of logit deformations $\Delta\ell$ in the 272-D model under filtered context subsets. All conditions exhibit a low-dimensional control subspace (effective rank $\approx 8$). REAL and SHUF show nearly identical variance spectra, indicating that predictive usefulness arises from directional alignment rather than rank or magnitude.

| Condition | Samples | cumvar@1 | cumvar@2 | cumvar@10 | eff. rank |
|---|---|---|---|---|---|
| REAL (helpful) | 2791 | 0.3118 | 0.3638 | 0.5275 | 8.28 |
| SHUF (on helpful) | 2791 | 0.3206 | 0.3768 | 0.5443 | 8.13 |
| SHUF (harmful) | 2108 | 0.3181 | 0.3754 | 0.5473 | 8.29 |

## 5. Limitations

This study has several important limitations.

First, we rely on simulated multi-region brain-state vectors paired with text contexts. This choice allows us to isolate mechanistic questions about controllability and evaluation, but it does not demonstrate that the observed effects will transfer directly to real neural recordings. Real brain signals are noisier, nonstationary, and subject to subject-specific variability, which may reduce the strength or reliability of control. Validating these findings on invasive or non-invasive neural datasets is a critical direction for future work.

Second, our evaluation focuses on token-level distributional metrics-Jensen-Shannon divergence, $\Delta$NLL, top-1 agreement, and logit-space geometry-rather than downstream generation quality or human judgments. While this choice is intentional, as it enables precise mechanistic analysis, it does not address whether brain conditioning improves perceived fluency, coherence, or usefulness of generated text. Bridging token-level control with sequence-level or human-evaluated outcomes remains an open challenge.

Third, although our control experiments rule out trivial perturbation explanations, they do not identify which specific dimensions or regions of the brain state drive particular semantic effects. Our analyses characterize deformation magnitude, token sensitivity, and subspace geometry, but stop short of attributing causal semantic roles to individual brain components. More fine-grained causal interventions-such as region-wise ablations, targeted projections, or counterfactual brain-state manipulations-would be needed to establish such mappings.

Fourth, our findings are limited to a specific architectural choice for brain injection, namely a learned brain-to-hidden projection in a Transformer language model. Different conditioning mechanisms, injection points, or model families may yield different control geometries or scaling behavior. Exploring how architectural choices shape the controllability and alignment of external signals is an important avenue for future research.

Finally, we emphasize that this work does not claim human-level semantic grounding or cognitive fidelity. The observed effects demonstrate structured control over a language model's output distribution, not equivalence to human thought or understanding. Interpreting brain-conditioned language models as cognitive models should therefore be done with caution.

## 6. Related Work

**Neural decoding of language and speech.** A long line of brain-computer interface (BCI) work shows that neural activity contains decodable information about intended communication, including handwriting and speech. Intracortical recordings have enabled high-rate text decoding from attempted handwriting (Willett et al., 2021; Makin et al., 2020), while ECoG and intracortical approaches have enabled speech decoding and even speech synthesis (Anumanchipalli et al., 2019; Moses et al., 2021; Willett et al., 2023). More recently, non-invasive recordings have been used to reconstruct aspects of continuous language semantics from fMRI (Tang et al., 2023). Our work is not focused on decoding natural language from real neural recordings; instead, we study a mechanistic question: whether a latent "brain-state" vector can act as a structured *control input* that predictably modulates next-token distributions.

**Conditioning and control in generative models.** Controllability has been explored extensively in text generation. Early approaches include conditional language modeling and attribute-controlled generation (Keskar et al., 2019; Dathathri et al., 2020). In diffusion models, classifier guidance and related techniques provide continuous "control knobs" over generation (Dhariwal & Nichol, 2021). Our alpha-scaling experiments are conceptually closest to these ideas: we treat brain state as a continuous control input and probe dose-response behavior in both distributional deformation (JS) and predictive utility ($\Delta$NLL).

**Injecting control signals into Transformers.** A practical design choice is how to inject external signals into Transformer-based models. The standard Transformer architecture (Vaswani et al., 2017) has inspired many conditioning mechanisms, including feature-wise modulation layers (Perez et al., 2018) and parameter-efficient tuning methods such as adapters and low-rank updates (Houlsby et al., 2019;

Hu et al., 2021). Prompt-based conditioning methods can be seen as learning continuous control vectors operating through the attention stack (Li & Liang, 2021; Lester et al., 2021). Brain conditioning in our setting is implemented via a learned projection pathway ("brain-to-hidden" injection), which is structurally analogous to these lightweight conditioning approaches but uses a time-varying latent state rather than a task prompt.

**Measuring influence beyond subjective generation quality.** A key difficulty in controllable generation is separating *change* from *useful change*. Many perturbations can induce comparable distribution shifts without improving the likelihood of the correct continuation. This motivates evaluation that jointly measures deformation (e.g., divergence from a baseline distribution) and task-aligned utility (e.g., NLL on the ground-truth token), similar in spirit to comparing controlled generation methods by both controllability and quality (Dathathri et al., 2020; Keskar et al., 2019). Our control suite (REAL vs. SHUF/GAUSS, plus alpha scaling) operationalizes this distinction: JS divergence quantifies the magnitude of deformation, while $\Delta$NLL tests whether the deformation moves probability mass toward the correct token.

**Control subspaces and low-rank structure.** Several empirical studies in modern deep networks suggest that meaningful interventions often occupy lower-dimensional subspaces of activation or output space, motivating analyses that probe low-rank structure of changes induced by conditioning or fine-tuning. This aligns with the hypothesis that brain conditioning may act through a compact "control subspace" in logit space, which we probe via PCA on $\Delta$logits (REAL vs. SHUF). Low-rank parameterizations in fine-tuning (Hu et al., 2021) further support the plausibility that strong control can emerge through a limited set of directions rather than uniform perturbations.

## 7. Conclusion

We investigated a mechanistic question about controllable language generation: whether a latent brain-state vector can act as a structured control input that predictably modulates a language model's next-token distribution. By treating brain activity as an exogenous conditioning signal rather than a decoding target, we framed brain-conditioned language modeling as a problem of probabilistic control. Across multiple analyses, we showed that brain input induces systematic, graded changes in token probabilities that go beyond generic perturbation and depend on alignment between neural state and linguistic context.

More broadly, this work introduces a quantitative framework for studying neural-state control of language models

that explicitly separates *deformation* of the output distribution from *useful* deformation aligned with correct prediction. By grounding evaluation in next-token distributions, divergence measures, and logit-space geometry, we provide tools for analyzing controllability independently of subjective generation quality. This perspective positions brain-state conditioning as an interpretable control problem in logit space and offers a foundation for future work on neural control signals, causal interventions, and validation on real brain data.

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
