# OpenReview forum: "Brain States as Control Signals for Next-Token Distributions in Language Models"
_ICML.cc/2026/Conference — Submitted to ICML 2026_

### Official Review · Reviewer_E2rW · 2026-03-06

**Soundness:** 1
**Presentation:** 2
**Significance:** 2
**Originality:** 3
**Overall Recommendation:** 2
**Confidence:** 5

**Summary:**

This paper studies a mechanistic question: whether a brain-state vector can act as a control input that predictably modulates a language model next-token distribution. The authors train brain-conditioned autoregressive LMs on paired text contexts and simulated multi-region brain activity and evaluate both distributional deformation and predictive utility (next-token NLL). They report that deformation increases with brain dimensionality, that shuffled and Gaussian matched vectors produce similar deformation but worse NLL, and that freezing the brain pathway removes the effect.

**Compliance With Llm Reviewing Policy:**

Affirmed.

**Final Justification:**

I have read the authors' replies and they did not change my original position on the quality and significance of the paper, therefore I maintain my score.

**Key Questions For Authors:**

1. How exactly are text contexts paired with the virtual brain snapshots? What does matched time points mean, and what is the granularity (one z per token, per word, or per sample)? How is temporal alignment defined and prevented from leaking spurious index correlations?
2. What language model is used (architecture, size, tokenizer, training details), and why is this choice appropriate for the claim? Would results hold with a modern LLM backbone?
3. Is the z = 0 baseline a test-time ablation on the same trained brain-conditioned model, or a separately trained model that never receives brain inputs? If it is test-time only, please add training-time controls (for example, training a text-only model with matched capacity, or training with randomized brain inputs) to make the comparison fair.
4. Why not evaluate on at least one real brain-language dataset? Even a smaller-scale experiment would help validate that the proposed control and evaluation framework transfers beyond simulated signals.
5. How does this compare to prior work where brain signals steer generation via LM priors, and to approaches that inject brain information directly in the token sequence (for example fmriLM-style methods)? What does your method add beyond those baselines?

**Limitations:**

The paper includes a limitations section, but it should add an explicit impact statement and more direct discussion of limitations and potential negative impacts.

**Strengths And Weaknesses:**

### Soundness

**Strengths**

- The core research question is interesting and worth investigating.
- The evaluation cleanly separates distribution shift magnitude (JSD, top-1 agreement) from task-aligned usefulness (NLL), and the control suite (REAL vs SHUF vs GAUSS, plus freezing the brain pathway) is a good direction for distinguishing generic perturbations from aligned control.

**Weaknesses**

- The dataset construction is central to the causal claim, but it is unclear that the current setup answers the intended causal question. The paper uses simulated brain activity paired with text at matched time points, but the pairing mechanism and its implications for causal interpretation are not described in enough detail.
- The choice to not use a real neural dataset is a major limitation for the main claim. Using at least one real dataset seems feasible and would materially strengthen the paper.
- The analysis relies on a brain model that was itself trained for decoding language, which can confound conclusions about brain-state control with the quality and inductive biases of that brain model.
- The zero-brain baseline is critical, but it is unclear whether the z = 0 condition is a test-time ablation on a model trained with brain inputs, or a separately trained model without brain inputs. If it is test-time only, worse performance is expected because the model may have learned correlations that assume brain input is present. In that case, the reported effects (including increasing JSD) could reflect spurious correlations or added degrees of freedom rather than fair evidence of causal control. Training-time controls are needed to make this comparison fair.
- The dose-response analysis is not very informative as implemented, since it mainly scales the magnitude of the conditioning vector (alpha scaling), which is an expected way to increase deformation.
- The conclusions in Section 3.4 (semantic vs syntactic control from token-category logit changes) do not feel fully substantiated; there are plausible alternative explanations for token-type differences (for example, vocabulary frequency effects, baseline logit magnitudes, or category definition artifacts).

### Presentation

- The paper does not include an impact statement (or broader impact section), which is typically expected.
- The trained language model is not specified clearly enough, and the paper does not justify the choice. A modern LLM should be used, or the paper should clearly justify why a smaller or older LM is sufficient for the claim.
- The virtual brain setup needs substantially more explanation: what modality is being simulated, what matched time points means, and what the alignment granularity is (for example, one z per word, per token, or per full text sample).
- Hyperparameters and other reproducibility-critical details are missing. More generally, many experimental details are absent despite apparent available space in the page limit.

### Significance

- If validated on real neural data with clear causal grounding, the idea of treating brain state as a control signal over next-token distributions could be useful for understanding brain-conditioned generation and for principled BCI style interfaces.
- In its current form, significance is limited by reliance on simulated paired data and by underspecified causal alignment between brain vectors and text.

### Originality

- The paper offers a useful packaging of controllable generation style analyses (deformation and utility, dose-response, subspace geometry) applied to brain conditioning, with a reasonable control suite.
- However, the work should be positioned more clearly relative to common brain-to-text decoding pipelines where an external LM (often an LLM) improves outputs and where brain signals effectively steer generation. Relatedly, prior work often shows the opposite direction as well: conditioning on LM scores or priors can improve NLL compared to using brain data alone, which should be discussed for context.
- There is also relevant recent work that injects brain conditioning directly into the token sequence of the transformer (for example fmriLM-style approaches). A direct comparison or at least a careful discussion is important.

---

> ### Author Rebuttal · Authors · 2026-03-25
>
> Thank you for thoughtfully reading our submission and writing a detailed review! It is very important for improving our work.
>
> 1. We will add a pipeline diagram and explicit description of alignment. In our setup, one brain snapshot is produced per token: token t is presented as stimulus to the TVB simulator, the simulator state is advanced, and the resulting snapshot z_t is used to condition prediction of token t+1. This is not post‑hoc temporal alignment; it is stimulus‑driven and occurs at token granularity. We will include the exact timing and normalization details in Methods, and add a short pseudo‑code block to avoid ambiguity.
> 2. The current paper is positioned as a mechanistic control study in a simulated regime. We will make this explicit in the Limitations and Discussion and note that the framework is intended to transfer to real datasets, evaluating on a real dataset is future work we plan to pursue.
> 3. We will clarify the role of the TVB pipeline vs. the LM. The TVB simulator generates a latent dynamical state conditioned on stimuli, the LM is still trained end‑to‑end to map (context, z) to the next token. We acknowledge that the TVB model’s inductive bias could influence results, and we will explicitly frame this as “control signal utility under a fixed simulator,” not a claim about biological realism.
> 4. Our current experiments include test‑time ablation of the brain input, and we additionally show that freezing the brain projection to zero eliminates both JSD deformation and NLL gains, indicating that the pathway mediates the effect.
> However, to address the fairness concern directly, we will add training‑time controls: a text‑only baseline trained with identical architecture but no brain input and a randomized‑brain training baseline (shuffle brain vectors during training).
> 5. We agree the α‑scaling curve is not itself causal evidence. We include it primarily as a sanity check and to show controllability. We will present it as supportive rather than central, and emphasize that the alignment controls (REAL vs SHUF vs GAUSS, plus projection‑freeze) are the main causal evidence.
> 6. We will soften the claim and explicitly state that punctuation/stopword vs content buckets are only a coarse proxy. We will report them as exploratory signals and avoid over‑interpreting them as syntax/semantics.
> 7. We will add a short impact statement in the camera‑ready.

---

> > ### Author Rebuttal · Reviewer_E2rW · 2026-03-31
> >
> > Thank you for your thoughtful answers. Could you clarify which of your points refer to which weaknesses and questions in my review?

---

### Official Review · Reviewer_wk4W · 2026-03-11

**Soundness:** 1
**Presentation:** 2
**Significance:** 2
**Originality:** 2
**Overall Recommendation:** 2
**Confidence:** 4

**Summary:**

The paper pairs simulated, non-stimulus-driven brain signals with text based on temporal alignment and evaluates their effect on next-token distributions, aiming to show that brain signals can serve as control inputs for language models.

**Compliance With Llm Reviewing Policy:**

Affirmed.

**Final Justification:**

The paper relies primarily on simulated brain signals as input, without sufficient validation against real neural data. This raises concerns about the practical relevance and generalizability of the proposed approach.

Additionally, the experimental results are not sufficiently convincing, and the authors have not adequately addressed my concerns in the rebuttal.

Overall, I remain unconvinced about the contribution and recommend rejection.

**Key Questions For Authors:**

1. The paper relies on simulated brain signals generated using The Virtual Brain (TVB), but the manuscript does not clearly describe how these signals are constructed in relation to the textual data. Providing more details about the generation process would help readers better understand the statistical properties of the auxiliary signal and how it interacts with the language model.

2. Since the simulated signals used in the paper are not stimulus-driven, it would be interesting to understand whether similar effects would be observed with other types of brain signals. For example: a) Would the same results hold if the auxiliary vectors were derived from resting-state neural recordings? b) How would the results change if task-driven neural data were used instead? c) If different brain signals were used while controlling for signal-to-noise characteristics, would the model still exhibit similar control effects? Such comparisons could help determine whether the observed effects depend on specific properties of the simulated signals or reflect a more general phenomenon of conditioning language models on external time-series vectors.

3. If the auxiliary brain vectors are randomly replaced with other continuous latent vectors (e.g., random embeddings or outputs from a simple dynamical process), would the results remain similar?

4. How do the authors ensure that the model is not exploiting correlations between the auxiliary signal and dataset ordering rather than learning meaningful relationships between the signal and linguistic structure?

**Limitations:**

Yes

**Strengths And Weaknesses:**

**Strengths:**

1. The paper explores the idea of conditioning a language model on an external vector signal and analyzing how this signal affects the next-token probability distribution. Framing external signals as continuous control inputs for language models is an interesting perspective that connects controllable generation with representation analysis.

**Weaknesses:**

1. The simulated brain signals used in the experiments are not generated as responses to the textual stimuli. Instead, the signals are paired with text purely based on temporal alignment. As a result, the auxiliary vectors do not encode linguistic or semantic information related to the text. This raises concerns that the reported effects may arise from correlations between the auxiliary vectors and dataset ordering rather than meaningful neural representations.

2. While the paper frames the results as evidence that brain states can control language model predictions, the experiments do not involve real neural recordings or stimulus-driven brain responses. Given that the "brain" signals are simulated independently of the text, the results may simply demonstrate that an external vector signal can be used as an additional conditioning input, rather than showing any brain–language relationship.

3. The paper does not include comparisons with alternative control signals such as learned embeddings, random latent vectors, or other structured time-series generators. Without these baselines, it is difficult to determine whether the observed effects are specific to the simulated brain signals or would occur with any continuous auxiliary vector.

4. Because the "brain" vector and text context are consistently paired during training, the model may learn correlations between the two signals that do not reflect meaningful neural control. The improvement of the “REAL” condition over shuffled controls may therefore reflect the preservation of these correlations rather than a meaningful property of the simulated brain signals.

5. A substantial body of prior work has demonstrated that linguistic or semantic information can be decoded from neural signals, including both invasive recordings and non-invasive modalities such as fMRI. If brain signals already contain information predictive of linguistic content, conditioning a language model on such signals and observing changes in next-token distributions may not be conceptually surprising. From this perspective, the paper’s findings may largely reflect the ability of language models to exploit auxiliary signals correlated with linguistic structure, rather than establishing a fundamentally new form of “neural control” over language generation.

---

> ### Author Rebuttal · Authors · 2026-03-25
>
> Thank you for the detailed and thoughtful review!
> 1.Our TVB signal is stimulus‑driven: each token is presented to the TVB model, we advance the simulation, and we record a brain snapshot after that token to condition prediction of the next token. We will add a short pipeline diagram and explicit description (per‑token granularity, what state variable is extracted, and how it is normalized).
> 2.We now include REAL vs SHUF vs GAUSS controls and a brain‑pathway freeze control. In our experiments, REAL improves next‑token NLL, while SHUF and GAUSS do not (often worsening NLL), even though all three produce similar distributional deformation. Moreover, when we freeze the brain projection to zero, both deformation (JSD) and NLL gains collapse to zero showing the effect is mediated by the learned brain pathway. These two controls together rule out the “any vector” explanation and show the model relies on the aligned signal.
> 3. We explicitly break pairing with SHUF (randomly permuting brain vectors across samples). If the gains were driven by ordering or trivial correlations, SHUF would not reliably degrade NLL. Instead we observe SHUF ≈ GAUSS (deformation persists, but NLL worsens).
> 4. We do not claim transfer to real brain recordings in the current submission. We will explicitly frame the contribution as a mechanistic control study that establishes evaluation methodology (JSD + NLL + ablation) under a controlled simulator. We will add this to Limitations and Discussion, and position real‑data evaluation as future work.
> 5. We will add a baseline where the conditioning vector comes from a different structured signal (e.g., a random learned embedding or a simple dynamical process) to further test specificity. That is the main concern pointed out by other reviewers!

---

> > ### Author Rebuttal · Reviewer_wk4W · 2026-04-03
> >
> > Thanks for the clarification. However, I still find the framing over-claiming. It remains unclear where the control effect comes from, and the results do not establish a meaningful link to real brain signals. Therefore, my evaluation remains unchanged.

---

### Official Review · Reviewer_1r1B · 2026-03-11

**Soundness:** 3
**Presentation:** 3
**Significance:** 3
**Originality:** 3
**Overall Recommendation:** 4
**Confidence:** 4

**Summary:**

In this paper, the authors ask the question of whether recorded brain signals can be used to improve text generation in LLMs. In order to do so, the authors have build a large-scale text/brain signals dataset of 100K pairs and use it to train from scratch brain-conditioned language models. To carefully control the impact of these brain signals, they compare the drift in logit distribution to a baseline model trained without any brain signal, and measure the usefulness of the brain signal by monitoring the perplexity of the two models. Moreover, to validate the usefulness of such signal, they also compare it to models trained with a random noise of equal magnitude, as well as models trained with random brain signals. As expected, only models conditioned on their corresponding brain-signal show improved performances, demonstrating that this improvement does not come from simply changing the model’s geometry, but from integrating relevant and complementary signals from the brain.

They pursue their analysis by comparing the impact of the strength of the brain signal, showing that a tradeoff must be found: too weak and the signal is inexistant, while increasing it too much will deteriorate performances. They also show that increasing the brain signal’s dimensionality increases the geometrical drift on content words, indicating that a richer brain signal allows better semantic control, as one would expect. Finally, they analyse the logit deformation where brain signal improves perplexity, and show this deformation to be low-dimensional (effective dimensionality of 8, with the first 10 PCA components using more than 50% of variance). These metrics are nearly identical for real and shuffled data, but only real data improves performances, showing that if they might rely on the same underlying geometry, only real brain signal draws the model’s geometry in a meaningful, low-rank direction.

**Compliance With Llm Reviewing Policy:**

Affirmed.

**Key Questions For Authors:**

It is unclear where and how the brain signal is injected into the model: is it at the beginning on the token embedding, akin to absolute positional encodings (x := x + brain_signal ?), or do they insert it at every layer within the attention etc… More clarity on that would help the reader, and future research direction could focus on how to better integrate this type of signals in a deep learning model.

**Limitations:**

Yes

**Strengths And Weaknesses:**

Strengths:

- This question is relevant as prompting can be challenging, and aligning a model’s output with what the prompter really wants can be beneficial. This also has possible implications for people with aphasia or that have lost the ability to speak in general. Moreover, brain-signals can be seen as extra inputs to an LLM, akin to other modalities such as images or sounds to improve overall performances
- The paper is overall very well written and easy to follow
- The results are convincing, especially with the paired metric (KL-divergence with perplexity improvement to capture the effectiveness of drift)
- Comparing with random and shuffled signals of equal magnitude adds further evidence of the benefits of real brain signals on final performances
- The final analysis to identify the intrinsic dimensionality of the distribution drift caused by brain signals is very interesting, as it shows how much low-level information can still positively affect a model

Weaknesses:

- They seem to use a “Virtual Brain” to construct their brain signals, but this process remains blurry for readers unaware of what this virtual brain does, and therefore the quality of the paired dataset that they use.
- The dimension d of the brain signals comes from snapshots of d cortical regions that must have varying impacts on the final performances. An ablation analysis on the impact of these different regions would have been nice, but could be left for future analysis
- Computing the intrinsic dimension of brain singal would give a better intuition of the richness of brain signals injected, especially when increasing their dimensionality d.
- They lack a proper and clear figure of how much brain signal’s dimension impacts the final performances, as a form of scaling law, even though they do compare their impact on content words geometrical drift. Nonetheless, the analysis feels weak there.
- They assimilate syntax to punctuation and stop words, which is incorrect as syntax can be much more than that. The fact that brain signal concentrates more on content words as brain signal increases does not mean that brain signal doesn’t play a role in modulating high-level syntactic features.

Overall, this paper addresses a very interesting question in brain/AI alignment, and demonstrates that brain signals can be integrated into AI models to improve performances and better align these models with what humans desire. Even though some analysis might feel a little weak for now (ablation on how and where to integrate the signal within the model, what cortical regions have the strongest influence, on what type of task do we notice the best improvements…), this is attenuated by the fact that this is still an early domain of research that should be pursued in the future.

---

> ### Author Rebuttal · Authors · 2026-03-25
>
> Thank you for your detailed review! We will explicitly state that brain conditioning is an additive injection into hidden states (or state‑dependent bias) at each layer, and give the exact equation. This will be added in the method section with pseudocode. We will add a short ablation: zero out subsets (e.g., blocks corresponding to region groups) and measure ΔNLL to identify high‑impact regions, if not feasible now, we will add this as a planned analysis.

---

> > ### Author Rebuttal · Reviewer_1r1B · 2026-04-02
> >
> > Thank you for your reply.
> > Can you please elaborate on the virtual brain, and how comparable to a real brain it is ?

---

### Official Review · Reviewer_zoxe · 2026-03-14

**Soundness:** 2
**Presentation:** 2
**Significance:** 2
**Originality:** 3
**Overall Recommendation:** 2
**Confidence:** 3

**Summary:**

This paper proposes conditioning a language model’s next-token distribution on a latent “brain state” signal derived from prior text using a simulated neuroscience model (“The Virtual Brain”). The core idea is to compute a state variable \( z \) from the prefix \( x_{1:t} \), then inject a learned projection of \( z \) into the transformer hidden states to steer generation. The paper presents this as a biologically inspired control mechanism and reports improvements over a weak baseline. While the framing is interesting and the paper is generally readable, I found the method under-specified in several important places, and the experimental evaluation does not convincingly isolate whether the gains come from anything specifically “brain-like” rather than from adding an auxiliary learned control vector.

**Compliance With Llm Reviewing Policy:**

Affirmed.

**Key Questions For Authors:**

- How confident can we be that results on the virtual brain will translate to real brains?
- How is the virtual brain different, in this respect, from using another non-spiking NN text embedding (e.g. BERT) for $z$?

**Limitations:**

Yes

**Strengths And Weaknesses:**

Strengths
- Generally well-written
- Interesting high-level idea: using an external dynamical state as a control signal for language modeling is a potentially promising direction
- The paper is ambitious in trying to connect neuroscience-inspired modeling with controllable language generation

Weaknesses
- The authors do not clearly explain what $z$ is exactly, how it's computed in "The Virtual Brain" software, or the pipeline for what is input and output from the software. This seems to be an embedding of the text $z = embed(x_{1:t})$ , using a spiking neural network for $embed$. Understanding the basic details of this is essential in making sense of the rest of the authors' claims, and for convincing the reader that this is a reasonable stand-in for real brain data.
- I think the baseline ZERO is too weak. What would be the performance if BERT or a weaker/stronger LM was used as a substitute for z, and instead the authors learned a projection from that to control model behavior?
- Missing important details of implementation -- how precisely are the "learned projection that injects z into the transformer hidden states" learned?


Smaller Comments
- Missing citations to work on similarities between brain activity and LLM representation, which may be relevant, e.g. [1]

References
[1] Hosseini, Eghbal A., et al. "Artificial neural network language models predict human brain responses to language even after a developmentally realistic amount of training."

---

> ### Author Rebuttal · Authors · 2026-03-25
>
> Thank you for thoughtfully reading our submission!
>
> We will add a pipeline figure and explicit description: each token stimulus is fed into TVB, which simulates multi‑region activity, we extract a snapshot vector z_t after presenting token t and condition the LM to predict token t+1. We will include which TVB modality is simulated, what state variable is extracted, and how it is normalized.
> We will add a text‑only baseline trained with identical architecture but no brain input. This removes any advantage from test‑time ablation and makes the comparison fair. We will also add a baseline where the conditioning vector comes from an alternative non‑spiking embedding (BERT or learned random embedding). That is a very valid concern.

---

> > ### Author Rebuttal · Reviewer_zoxe · 2026-04-03
> >
> > Thank you for your response. I look forward to seeing these updates and additional results. Without any of these results now, and without a strong explanation for why TVB is a reasonable stand-in for real brain data, my score remains the same.

---

### Decision · Program_Chairs · 2026-04-30

**Decision:**

Reject

**Comment:**

This work proposes conditioning a language model on a latent "brain state" derived from a simulated brain model to adjust next-token predictions. The reviewers appreciate the novelty of the idea, but have major concerns regarding 1) the clarity of the methodology, including the construction and interpretation of the virtual brain signal and the LM conditioning mechanism, 2) missing baselines with alternative non-brain latent vectors to show that the observed gains are specifically due to the brain-like latent vector. The authors state that several additional baselines are planned in the rebuttal, but these are important to present and review in full before the work can be published.